# Methods of Analyzing the Error and Rectifying the Calibration of a Solar Tracking System for High-Precision Solar Tracking in Orbit

Yingqiu Shao *, Zhanfeng Li, Xiaohu Yang, Yu Huang, Bo Li, Guanyu Lin and Jifeng Li

Changchun Institute of Optics, Fine Mechanics and Physics, Chinese Academy of Sciences, Changchun 130033, China; lizhanfeng@ciomp.ac.cn (Z.L.); yangxiaohu@ciomp.ac.cn (X.Y.); huangyu@ciomp.ac.cn (Y.H.); libo01@ciomp.ac.cn (B.L.); linguanyu@ciomp.ac.cn (G.L.); lijifeng@ciomp.ac.cn (J.L.)
* Correspondence: shaoyq@ciomp.ac.cn; Tel.: +86-0431-8670-8136

**Abstract:** Reliability is the most critical characteristic of space missions, for example in capturing and tracking moving targets. To this end, two methods are designed to track sunlight using solar remote-sensing instruments (SRSIs). The primary method is to use the offset angles of the guide mirror for closed-loop tracking, while the alternative method is to use the sunlight angles, calculated from the satellite attitude, solar vector, and mechanical installation correction parameters, for open-loop tracking. By comprehensively analyzing the error and rectifying the calibration of the solar tracking system, we demonstrate that the absolute value of the azimuth tracking precision is less than $0.0121°$ and the pitch is less than $0.0037°$ with the primary method. Correspondingly, they are $0.0992°$ and $0.0960°$ with the alternative method. These precisions meet the requirements of SRSIs. In addition, recalibration due to mechanical vibration during the satellite's launch may invalidate the above methods, leading to the failure of SRSIs. Hence, we propose a dedicated injection parameter strategy to rectify the sunlight angles to capture and track the sunlight successfully. The stable and effective results in the ultraviolet to near-infrared spectrum validate the SRSI's high-precision sunlight tracking performance. Furthermore, the above methods can also be applied to all orbital inclinations and may provide a solution for capturing and tracking moving targets.

**Keywords:** solar remote-sensing instruments; sunlight tracking; high precision; solar tracking system





## 1. Introduction

The earth is extremely vulnerable to changes in solar radiation; these small changes affect energy, water, carbon, and nitrogen cycle processes, in turn causing global changes to the environment and climate [1–10]. Therefore, monitoring such changes in solar radiation can provide important scientific data for long-term research on changes in the earth's climate and atmospheric composition, as well as solar physics. Since the 1970s, many SRSIs have been developed, including the Solar and Heliospheric Observatory (SOHO) jointly developed by the European Space Agency (ESA) and the National Aeronautics and Space Administration (NASA) [11]. The beginning of the 21st century saw the development of the Solar Variability Irradiance Monitor (SOVIM) [12], the Solar Spectrum (SOLSPEC) instrument [13], and a Solar Auto-Calibrating extreme-ultraviolet (EUV)/ultraviolet (UV) Spectrophotometer (SOL-ACES) [14,15] loaded on the International Space Station; the Solar Orbiter was jointly developed by NASA and ESA and launched in 2020.

Solar radiation detection research in China was initiated at the beginning of the 21st century. The solar ultraviolet monitor, developed by the Changchun Institute of Optics, Fine Mechanics and Physics, Chinese Academy of Sciences (CIOMP), launched on the Shenzhou III spacecraft in 2002 and successfully obtained a solar ultraviolet spectral radiation signal; this was China's first scientific and technological experimental satellite for solar exploration.

There are three international schemes for sunlight tracking using SRSIs: satellite tracking, fixed-point tracking at the Sun–Earth Lagrange point (L1), and special turntable tracking. The SOlar VAriability PICARD (SOVAP) on the PICARD satellite and Solar Radiation and Climate Experiment (SORCE) of the Total Irradiance Monitor (TIM) satellite are typical examples of the satellite-tracking scheme. Here, in order to control the satellite's attitude to ensure it is always aimed towards the sunlight, its tracking accuracy is less than 100″. This scheme is suitable for small, special solar observation satellites. The SOHO of the Variability of the Solar Irradiance and Gravity Oscillations (VIRGO) satellite is a typical example of fixed-point tracking; it is locked at the Sun–Earth Lagrange point (L1) and permanently aimed at the sun after launch. This scheme is used to adjust its orbit and maintain attitude, with the advantages of less energy usage, a long life, and a long solar observation time. The coarse pointing device (CPD) of the international space station (ISS) is a typical example of special turntable tracking. The two-dimensional turntable is controlled with a sun sensor to track sunlight. The tracking accuracy of the CPD is $\pm 1°$; the rotation range of the two axes is $\pm 40°$ and $\pm 25°$, respectively; and the effective observation time of the sun is 15 min. However, the ISS/CPD suffered from solar sensor, tracking, and power failure, and only completed an effective solar measurement for 1 month, with 8 months in orbit—less than the design life of 1.5 years [16–18].

China's Fengyun-3E satellite, which was launched in July 2021, is loaded with the CIOMP-developed solar irradiance spectrometer (SIS), which operates in the ultraviolet to near-infrared band (165~2400 nm) and accumulates a large quantity of scientific data on solar radiation changes. This satellite operates in a sun-synchronous orbit; its design life is 8 years, and it weighs 6 tons. It is not suitable for fixed-point tracking at the Sun–Earth Lagrange point (L1) or the satellite tracking scheme. The payloads on the satellite are only suitable for the special turntable tracking scheme.

Since the annual variation in solar spectral irradiance in the visible and near-infrared bands is only $10^{-3}$, the stability of the SIS is required to be less than 0.2%/year in orbit [19–23]. In order to meet this requirement, the solar tracking accuracy must be less than 0.1°. Two methods are designed for the SIS-based solar tracking in orbit: one method requires a solar sensor and the other does not. The absolute value of the azimuth tracking precision is less than 0.0121° and the pitch is less than 0.0037° with the primary method; correspondingly, they are 0.0992° and 0.0960° with the alternative method. Even if the sun sensor fails, the SIS can meet the tracking accuracy requirements. In addition, recalibration due to mechanical vibration during the satellite's launch may invalidate the above methods, leading to the failure of the SIS. Hence, we propose a dedicated injection parameter strategy to rectify the sunlight angles in order to capture and track the sunlight successfully. The sunlight angle can be corrected in orbit. Even if the vibration causes an offset between the mechanical structures during launch, it is ensured that the SIS can still capture the sunlight successfully in orbit. Furthermore, two methods for solar tracking and the SIS strategy can also be applied to all orbital inclinations and may provide a solution for SRSIs to capture and track moving targets successfully.

## 2. Materials and Methods

### 2.1. The Principle of Solar Tracking

The components of the spectrometer and the coordinate systems are illustrated in Figure 1a. The spectrometer is composed of a turntable electrical cabinet and a front detector. The orbital coordinate system, satellite coordinate system, turntable (mounting surface) coordinate system, axis of the turntable coordinate system, and guide-mirror coordinate system are theoretically coincident. The orbit diagram is illustrated in Figure 1b. The satellite platform moves along the +X axis direction, the sunlight is incident from the −Y axis direction, and the earth is in the +Z axis direction. The front detector's installation position is shown in Figure 1c. The front detector is installed on the XOY plane of the satellite in the −Z direction. The turntable's electrical cabinet is in the capsule of the satellite. A block diagram of the spectrometer's working principle is shown in Figure 2a.

The front detector consists of a spectrometer electrical cabinet, a spectrometer (including UV, VIS, and NIR channels), a guide mirror, a turntable azimuth motor and encoder, and a pitch motor and encoder. The electronics unit of the front detector is located on the turntable mechanism.

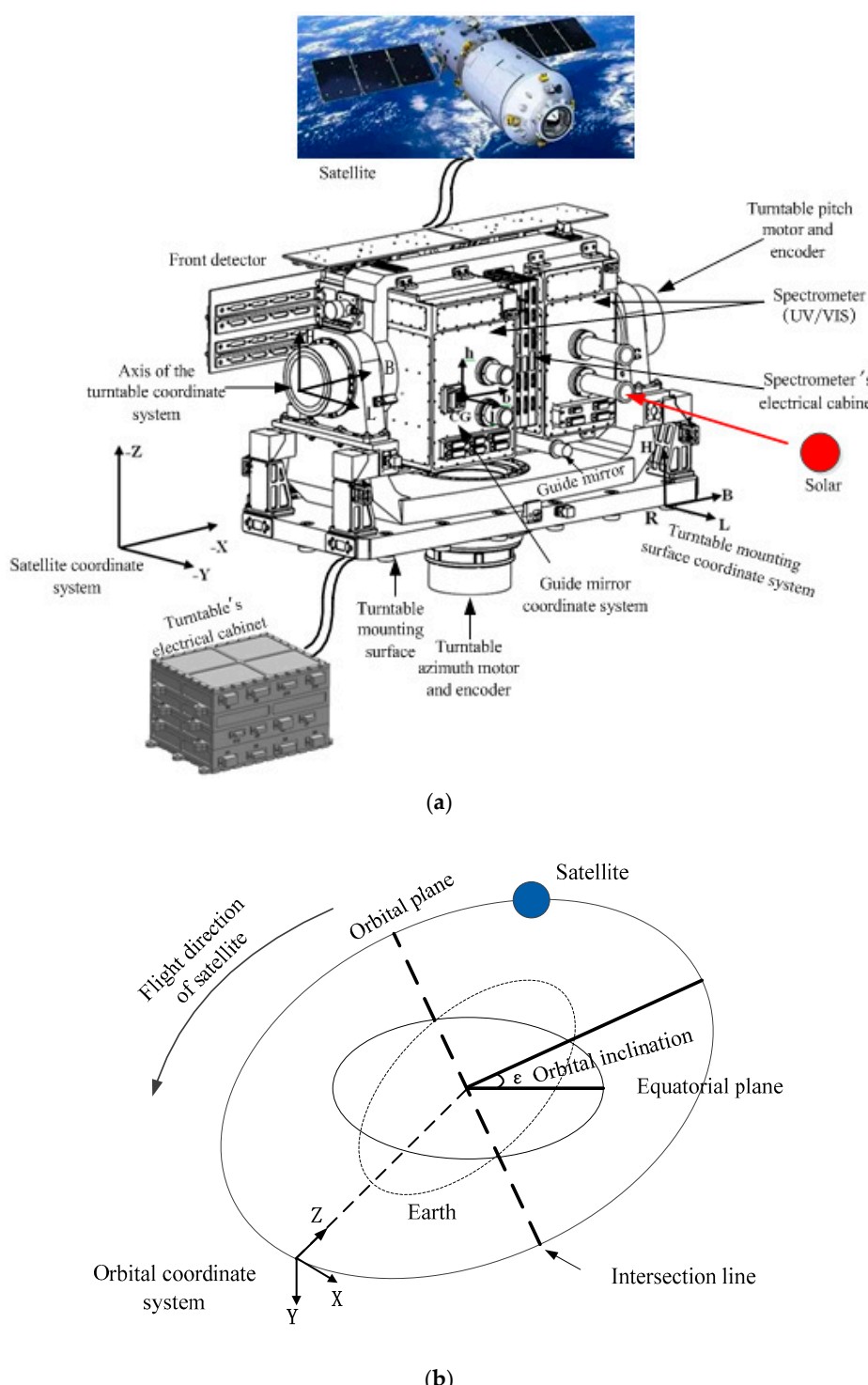

(**a**)

(**b**)

**Figure 1.** *Cont.*

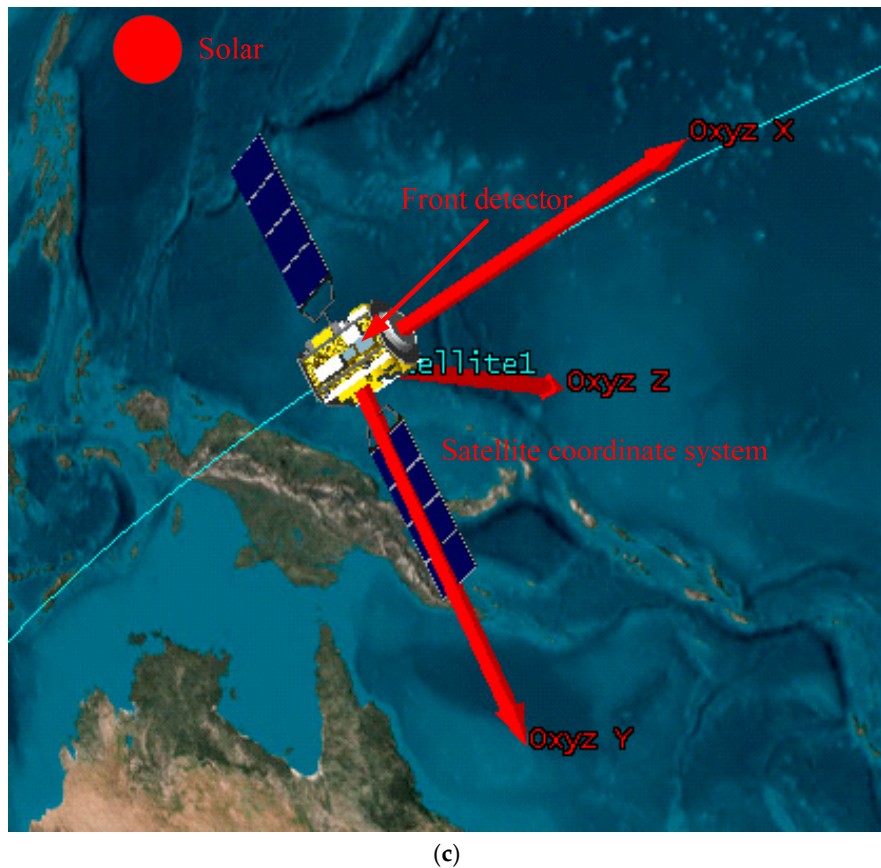

(**c**)

**Figure 1.** (**a**) The components of the spectrometer and the coordinate systems. (**b**) Orbit diagram. (**c**) The position of front detection on satellite.

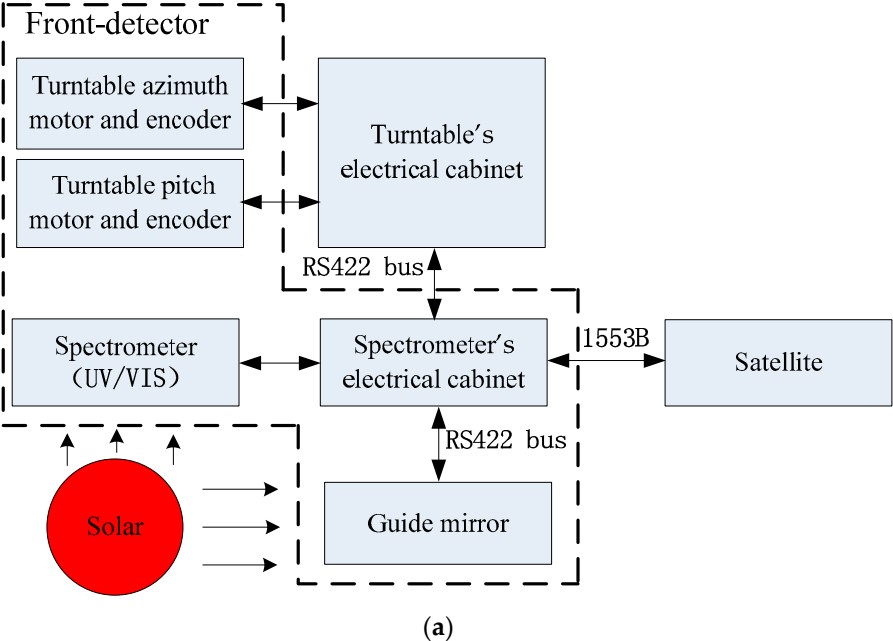

(**a**)

**Figure 2.** *Cont.*

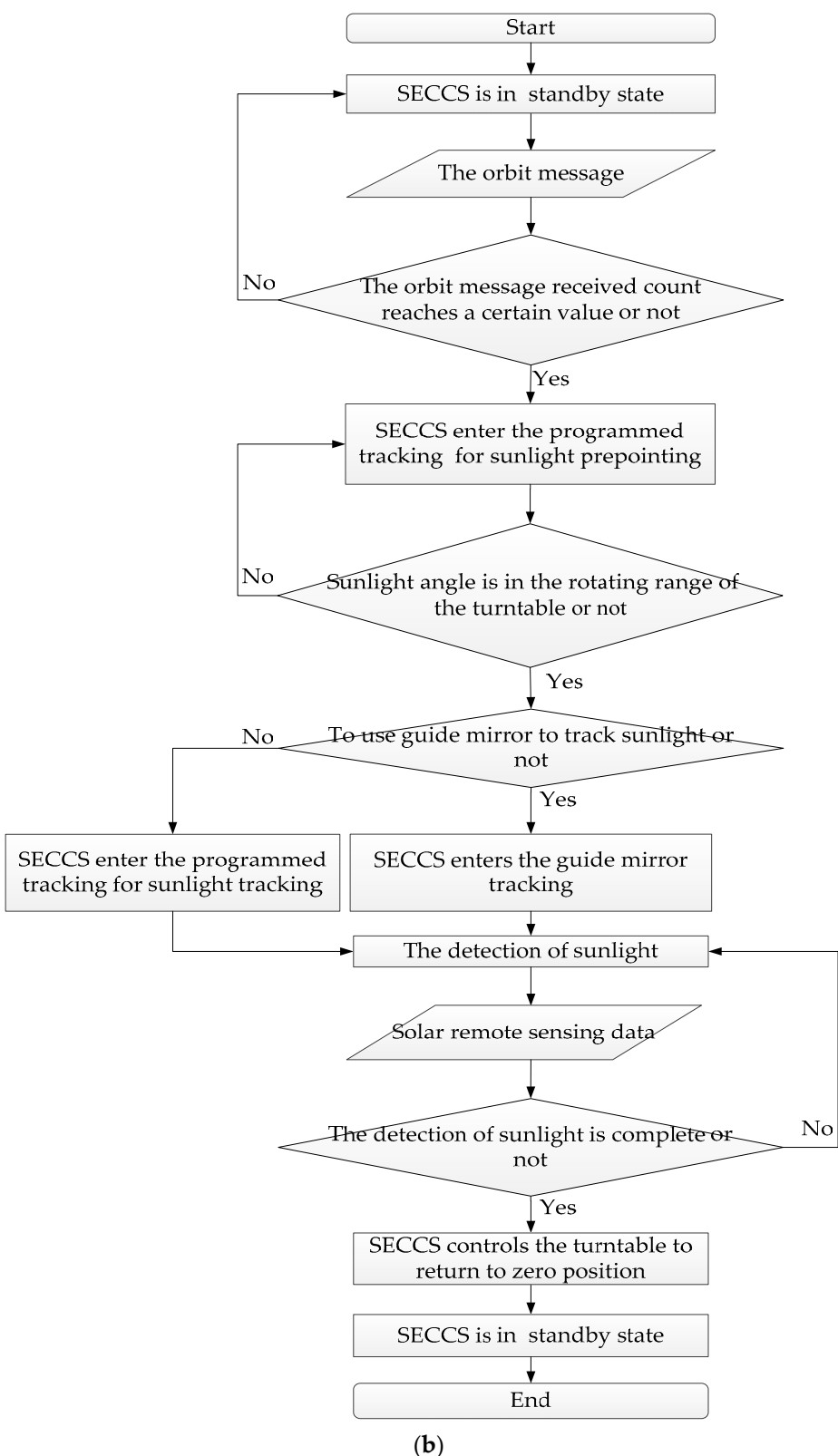

**Figure 2.** (**a**) Working-principle diagram of the instrument. (**b**) Turntable control flow.

The spectrometer's electrical cabinet receives the broadcast message (satellite attitude) and the orbit message (solar vector, etc.) sent by the satellite through the 1553B bus. With these data and the mechanical installation correction parameters, the azimuth and pitch angles of the solar radiation in the guide-mirror coordinate system are calculated by

software in real time. On one hand, these angles are used to measure the time it takes for the spectrometer's electrical cabinet to signal the spectrometer's electronics unit to power on and preheat; on the other hand, they are used to measure the time it takes for the same cabinet to signal the turntable, through the turntable's electrical cabinet, to capture the solar radiation in the guide mirror's field of view. The turntable's electrical cabinet receives the azimuth and pitch angles sent by the spectrometer's electrical cabinet through the RS422 bus and controls the corresponding rotation angle of the two-dimensional turntable. The azimuth and pitch encoders are used for the closed-loop position.

The turntable control flow is shown in Figure 2b. First, the spectrometer's electrical cabinet's CPU software tool (SECCS) keeps the turntable in the standby state. When the received orbit message count reaches a certain value, the working mode is switched to a programmed tracking mode (solar prepointing). At that time, the sunlight is outside the rotation range of the turntable, and the azimuth or pitch angle of the turntable is at the limit position. When the sunlight angle is in the turntable's rotating range, The SECCS decides to use either the primary or alternative method according to the injection. If the primary method is used to track the sunlight, the working mode is switched to the guide mirror tracking.

The primary method uses a closed-loop tracking of the guide mirror's offset; when the sunlight enters the rotating range of the turntable, the turntable is controlled to quickly direct the sunlight into the guide mirror's central field of view. Then, the turntable's electrical cabinet controls the turntable according to the solar offset angle at the guide mirror's central field of view (a $2° \times 2°$ circular field), initiating the closed-loop tracking. The azimuth and pitch offsets, sent by the guide mirror, are received by the spectrometer's electrical cabinet through the RS422 bus and sent to the turntable's electrical cabinet. When the sunlight is in the central field of view of the guide mirror within $0.1°$, it is observed by the spectrometer (controlled by the spectrometer's electrical cabinet), which then sends the remote sensing data to the satellite through the 1553B bus.

If the alternative method is used for daily tracking, the working mode is still the programmed tracking mode, wherein sunlight tracking is carried out. The sunlight azimuth and pitch angles calculated by the SECCS in the guide-mirror coordinate system are used for open-loop tracking; the turntable's electrical cabinet controls the turntable for solar tracking according to these angles, which are sent by the spectrometer's electrical cabinet. When the difference between the position of the corresponding encoder and the corresponding angle calculated by the SECCS is less than $0.05°$, the spectrometer is controlled by the spectrometer's electrical cabinet to observe the solar radiation and send the remote sensing data to the satellite through the 1553B bus. Meanwhile, the guide mirror's offset can be obtained to analyze the quality of solar tracking in orbit.

When the sunlight detection is complete, the SECCS controls the turntable to return to the zero position and enter the standby state. The spectrometer waits for the next sunlight measurement.

*2.2. Error Analysis and Rectifying the Deviation of the Solar Tracking System*

2.2.1. Analysis of the Errors and Deviations of the Solar Tracking System

1. The inherent errors and deviations of the solar tracking system include:

   (a) Error caused by software time delay.

   The error consists of two parts; firstly, the error caused by the time delay T1 that is from the solar angle calculated by the SECCS to be received by the turntable's electrical cabinet. Secondly, the error caused by the time delay T2 that is from the turntable's electrical cabinet receiving the azimuth and pitch angles to control the corresponding motor.

   (1) The maximum main cycle period of the SECCS is 0.3 s, so the maximum of T1 is 0.3 s, and the maximum speed of the solar movement is $0.035°$/s. Therefore, the maximum error generated by this time delay is $0.0105°$.

(2)    The main cycle of CPU software of the turntable's electrical cabinet is 1 ms, so the maximum of T2 is 1 ms, and the maximum error generated during this time is $0.000035°$ (negligible). It cannot be rectified but can be ignored.

(b)    Error caused by the SECCS calculation.
This part of the error only involves that which is generated by the SECCS data processing. The chip of the solar irradiance spectrometer's electrical cabinet's CPU is a DSP, which can calculate 64-bit floating-point operations. Therefore, the CPU calculation error is $10^{-6}$ rad (negligible). It cannot be rectified but can be ignored.

(c)    Error and deviation caused by mechanical installation.
This part of the error and deviation includes 4 items:

(1)    The installation deviation and measurement error between the turntable and the satellite's mounting surface are in Table 1 (no. 4).

(2)    The installation deviation between the turntable's mounting surface and the shaft system of the turntable and the measurement error are in Table 1 (no. 5).

(3)    The installation deviation and measurement error of the guide mirror's optical axis and the shaft system of the turntable are in Table 1 (no. 6).

(4)    The installation deviation and measurement error of the guide mirror's optical axis and vis channel optical axis are in Table 1 (no. 7).

(d)    Error caused by the turntable's tracking control. The maximum error of the control turntable is $0.01°$, and the error cannot be rectified.

(e)    The error in data provided by the satellite.

(1)    The solar vector error of the orbital coordinate system error is $0.009°$.

(2)    The error caused by the update time of the solar vector and satellite attitude is 40″.

(3)    The cycle time of the orbit message sent to the spectrometer by the satellite is 2 s, and the maximum speed of the solar movement is $0.035°/s$. Therefore, the maximum error generated by this cycle time is $0.07°$.

(f)    The satellite sends its attitude angle in a broadcast message to the spectrometer, and the attitude angle is $0.006°$ ($3\sigma$) in Table 1 (no. 9).

2.    The unpredictable deviations of the solar tracking system include:

(a)    The solar tracking simulation experiment on the ground is not sufficient, resulting in incorrect symbols of the angle values.

(b)    The solar tracking simulation experiment on the ground is not sufficient, resulting in an inaccurate angle calculation.

(c)    During satellite launch, new deviations occur between the mechanical structures due to vibration.

If the sunlight angle calculation is wrong or large deviations occur during launch, the assignment will fail. The sunlight angle calculation and rectification are highly important; the SECCS can rectify the sunlight angle value or not and modify the signs of this angle according to the injection parameters. The calculation and rectification of the sunlight angle are not affected by the orbital inclination in orbit.

In conclusion, the maximum inherent error of the solar tracking system is $0.118027°$, which is $0.08347°$ converted to the azimuth and pitch angle components. Therefore, the circular field of view of the guide mirror was designed to be $2° \times 2°$. This can ensure that the solar irradiance spectrometer can successfully capture solar changes. At the same time, the open-loop tracking using the solar angle calculated by the software tool can ensure that the tracking precision meets the requirements.

**Table 1.** The inherent errors and deviations of the solar tracking system.

| Serial No. | Error/Deviation Item | | Value of the Deviation | Value of the Error | Correction Plan |
|---|---|---|---|---|---|
| 1 | Error caused by software time delay | T1 | - | 0.0175° | Cannot be rectified |
| 2 | | T2 | - | 0.000035° | Can be ignored |
| 3 | Software calculation error | | - | $10^{-6}$ rad | Can be ignored |
| 4 | Mechanical installation deviation and measurement error (angle of rotation around Z/X/Y in order) | Between the turntable and the satellite's mounting surface | $-0.30012'/0.13335'/-0.38334'$ | 0.5″ | Rectified by the SECCS |
| 5 | | Between the turntable's mounting surface and the axis of the turntable coordinate system | $-1.30744'/2.60464'/-0.27004'$ | 0.5″ | Rectified by the SECCS |
| 6 | | Between the guide mirror and the axis of the turntable coordinate system | $1.05779'/-2.25331'/0.203515'$ | 0.5″ | Rectified by the SECCS |
| 7 | | Between the guide mirror and the vis channel optical axis | $0.95779'/-1.95331'/0.303515'$ | 0.5″ | Rectified by the guide mirror's CPU software |
| 8 | Error caused by the turntable's tracking control | | - | 0.01° | Cannot be rectified |
| 9 | Deviation | Satellite attitude | - | 0.006° | Rectified by the SECCS |
| 10 | Error in data provided by the satellite | The solar vector error of the orbital coordinate system | - | 0.009° | Cannot be rectified |
| 11 | | Error caused by the update time of the solar vector and the satellite attitude data | - | 40 t | Cannot be rectified |
| 12 | | Error caused by the cycle time of the orbit message | - | 0.07° | Cannot be rectified |

### 2.2.2. Rectification Method for the Deviation of the Solar Tracking System

The deviation of the solar vector, mechanical installation, etc., influences the azimuth and pitch angle values in the guide-mirror coordinate system. Therefore, the solar angle calculation process is the same as the process of correcting the solar tracking system's deviation. The SECCS transforms the solar vector in the orbital coordinate system into a solar vector in the guide-mirror coordinate system through a coordinate conversion; from the solar vector in the latter system, the SECCS calculates the solar azimuth and pitch angles. The sunlight angle calculation flow diagram is shown in Figure 3.

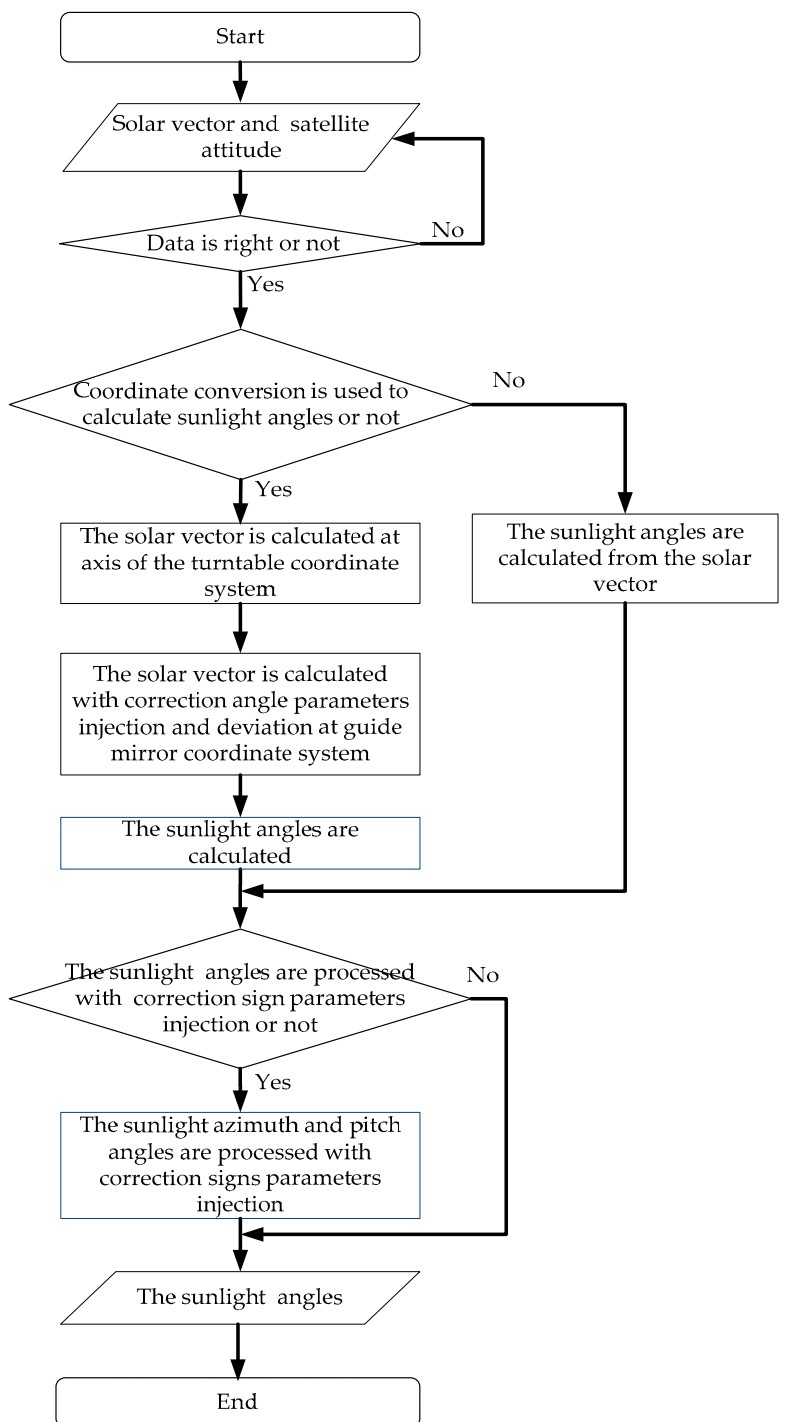

**Figure 3.** Angle calculation flow diagram.

During this calculation process, the involved coordinate systems include the orbital coordinate system, the satellite coordinate system, the turntable (mounting surface) coordinate system, the axis of the turntable coordinate system, and the guide-mirror coordinate system. The positional relationship between these coordinate systems is shown in Figure 1. Since the distance between the Earth and the Sun can be considered infinite, the solar angle value will not be affected by the translation of each coordinate system. The relationship between the coordinate systems after translation is shown in Figure 4a. The solar vector of the orbital coordinate system and the mechanical installation have inherent deviations. Therefore, the solar angle in the guide-mirror coordinate system is calculated from the solar vector in the orbital coordinate system. Four coordinate transformations are required to rectify the deviation. The transformation data flow diagram is shown in Figure 4b. In this process, the deviation of nos. 4, 5, 6, and 9 in Table 1 is rectified, and the installation deviation of no. 7 in Table 1 is rectified by the guide mirror's CPU software after mechanical installation and measurement.

The rotation order of the deviation no. 9 in Table 1 between the satellite coordinate system and the orbital coordinate system is around the $Z$ axis (yaw angle, $\psi_1$), the $X$ axis (rolling angle, $\varphi_1$), and the $Y$ axis (pitch angle, $\theta_1$), in turn. In other words, the satellite's attitude are $\psi_1$, $\varphi_1$, $\theta_1$, and the satellite sends its attitude parameter to the payload every second. The rotation of the coordinate axes between the coordinate systems is carried out in that order.

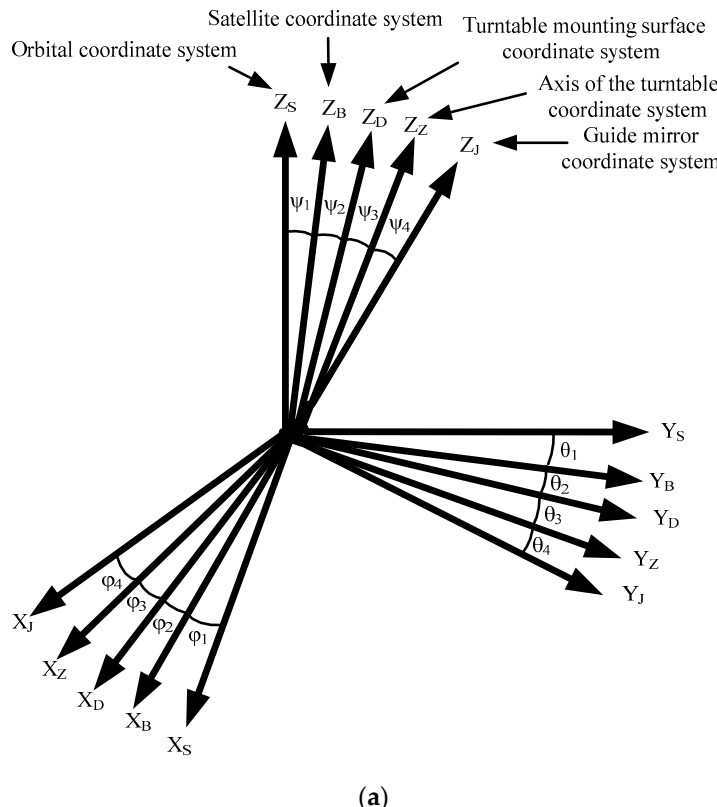

(**a**)

**Figure 4.** *Cont.*

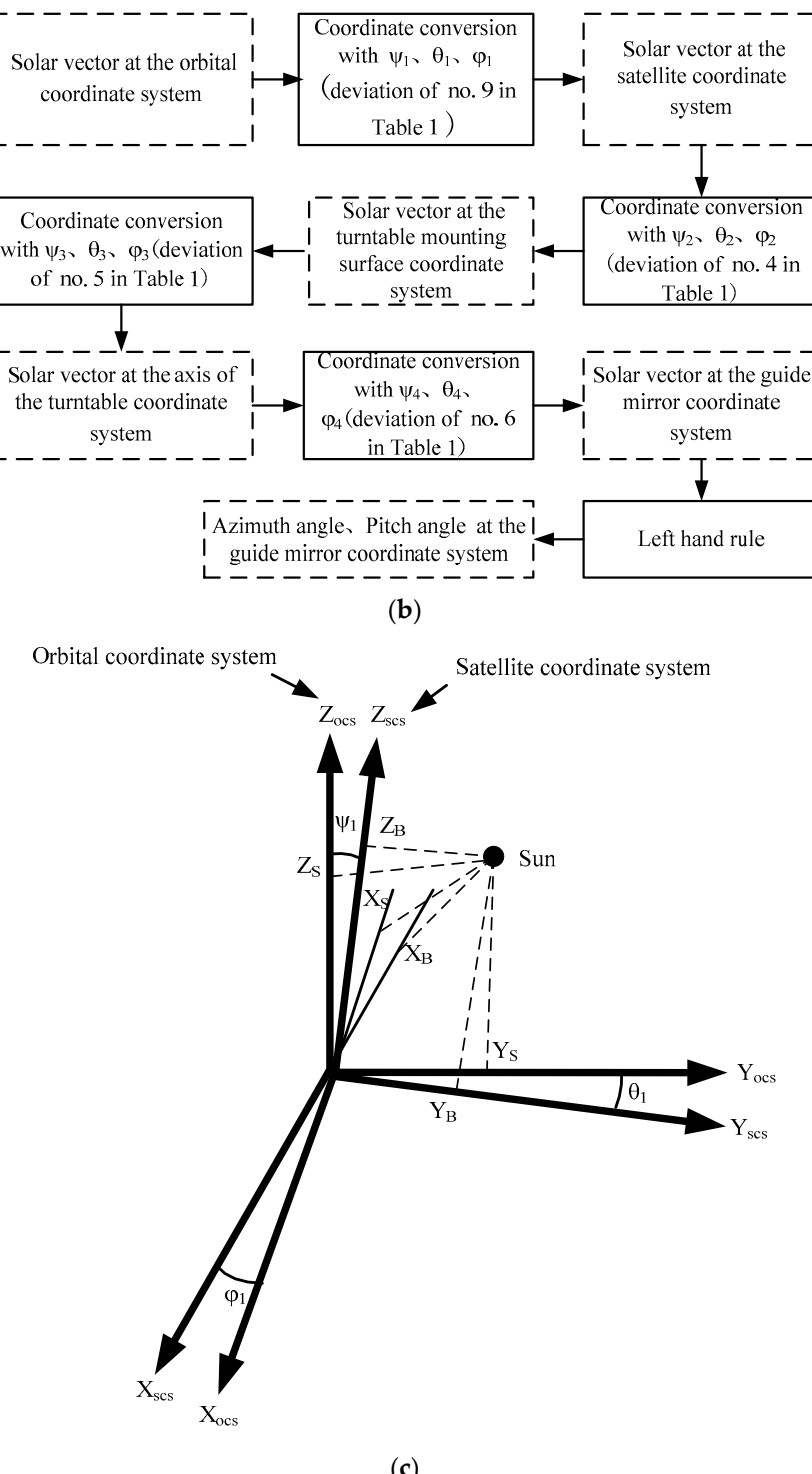

**(b)**

**(c)**

**Figure 4.** (**a**) Relationship between coordinate systems of the solar tracking system. (**b**) Coordinate transformation data flow diagram (see Table 1). (**c**) Diagram of the sun in the orbital coordinate system and the satellite coordinate system.

Without considering the three-axis, nonorthogonal error of each coordinate system, assume that the solar vector of the orbital system is $[X_S \ Y_S \ Z_S]$ meeting $(X_S/32767)^2 + (Y_S/32767)^2 + (Z_S/32767)^2 = 1$. The solar vector transformation from the orbital coordinate system to the satellite coordinate system is $[X_B \ Y_B \ Z_B]$, shown on a diagram of the sun in the orbital coordinate system and the satellite coordinate system in Figure 4c and reflected in Formula (1):

$$\begin{bmatrix} X_B \\ Y_B \\ Z_B \end{bmatrix} = \begin{bmatrix} \cos\psi_1\cos\theta_1 - \sin\psi_1\sin\phi_1\sin\theta_1 & \sin\psi_1\cos\theta_1 + \cos\psi_1\sin\phi_1\sin\theta_1 & -\cos\phi_1\sin\theta_1 \\ -\sin\psi_1\cos\phi_1 & \cos\psi_1\cos\phi_1 & \sin\phi_1 \\ \cos\psi_1\sin\theta_1 + \sin\psi_1\sin\phi_1\cos\theta_1 & \sin\psi_1\sin\theta_1 - \cos\psi_1\sin\phi_1\cos\theta_1 & \cos\phi_1\cos\theta_1 \end{bmatrix} \begin{bmatrix} X_S \\ Y_S \\ Z_S \end{bmatrix} \tag{1}$$

The deviation rotation angles around the $Z$, $X$, and $Y$ axes are $\psi_2$, $\varphi_2$, and $\theta_2$, respectively (no. 4 in Table 1). Then, the solar vector transformation from the satellite coordinate system to the turntable (mounting surface) coordinate system is [$X_D$ $Y_D$ $Z_D$], reflected in Formula (2):

$$\begin{bmatrix} X_D \\ Y_D \\ Z_D \end{bmatrix} = \begin{bmatrix} \cos\psi_2\cos\theta_2 - \sin\psi_2\sin\phi_2\sin\theta_2 & \sin\psi_2\cos\theta_2 + \cos\psi_2\sin\phi_2\sin\theta_2 & -\cos\phi_2\sin\theta_2 \\ -\sin\psi_2\cos\phi_2 & \cos\psi_2 & \cos\phi_2 & \sin\phi_2 \\ \cos\psi_2\sin\theta_2 + \sin\psi_2\sin\phi_2\cos\theta_2 & \sin\psi_2\sin\theta_2 - \cos\psi_2\sin\phi_2\cos\theta_2 & \cos\phi_2\cos\theta_2 \end{bmatrix} \begin{bmatrix} X_B \\ Y_B \\ Z_B \end{bmatrix} \tag{2}$$

The deviation rotation angles around the $Z$, $X$, and $Y$ axes are $\psi_3$, $\varphi_3$, and $\theta_3$, respectively (no. 5 in Table 1). Then, the solar vector transformation from the turntable's (mounting surface) coordinate system to the axis of the turntable coordinate system is [$X_Z$ $Y_Z$ $Z_Z$], reflected in Formula (3):

$$\begin{bmatrix} X_Z \\ Y_Z \\ Z_Z \end{bmatrix} = \begin{bmatrix} \cos\psi_3\cos\theta_3 - \sin\psi_3\sin\phi_3\sin\theta_3 & \sin\psi_3\cos\theta_3 + \cos\psi_3\sin\phi_3\sin\theta_3 & -\cos\phi_3\sin\theta_3 \\ -\sin\psi_3\cos\phi_3 & \cos\psi_3\cos\phi_3 & \sin\phi_3 \\ \cos\psi_3\sin\theta_3 + \sin\psi_3\sin\phi_3\cos\theta_3 & \sin\psi_3\sin\theta_3 - \cos\psi_3\sin\phi_3\cos\theta_3 & \cos\phi_3\cos\theta_3 \end{bmatrix} \begin{bmatrix} X_D \\ Y_D \\ Z_D \end{bmatrix} \tag{3}$$

Assume that the solar vector in the guide-mirror coordinate system is [$X_J$ $Y_J$ $Z_J$]. The deviation rotation angles around the $Z$, $X$, and $Y$ axes are $\psi_4$, $\varphi_4$, and $\theta_4$, respectively (no. 6 in Table 1).

During the satellite's launching process, its mechanical system and payload are affected by the vibration, causing a new angle to occur in the relative position between the coordinate systems. Since the satellite is already in orbit, the angle change value cannot be measured. Due to the guide mirror's small angle of view, if the new angle is too large (such as 1°), the solar irradiance spectrometer will be unable to capture the solar position, resulting in failure in orbit. The solar angle value calculated by software will not be accurate, which may lead to the same result. In order to solve this problem, the SECCS can receive correction parameters in orbit and superimpose them. According to the mechanical experiment results on the ground and to avoid any misinjection, the maximum correction parameter is 5°. If it exceeds 5°, the SECCS will not superimpose it.

Assume that the injection correction parameters of the $Z$, $X$, and $Y$ axes are $\psi_x$ (yaw angle), $\varphi_x$ (rolling angle), and $\theta_x$ (pitch angle), respectively. All of the default values of the correction parameters are 0°; then the solar vector [$X_Z$ $Y_Z$ $Z_Z$] at the axis of the turntable coordinate system is converted to the sun vector [$X_J$ $Y_J$ $Z_J$] in the guide-mirror coordinate system, as shown in Formula (4).

$$\begin{bmatrix} X_J \\ Y_J \\ Z_J \end{bmatrix} = \begin{bmatrix} \cos(\psi_4+\psi_x)\cos(\theta_4+\theta_x) - \sin(\psi_4+\psi_x)\sin(\phi_4+\phi_x)\sin(\theta_4+5ta_x) & \sin(\psi_4+\psi_x)\cos(\theta_4+\theta_x) + \cos(\psi_4+\psi_x)\sin(\phi_4+\phi_x)\sin(\theta_4+\theta_x) & -\cos(\phi_4+\phi_x)\sin(\theta_4+\theta_x) \\ -\sin(\psi_4+\psi_x) & \cos(\phi_4+\phi_x) & \cos(\psi_4+\psi_x)\cos(\phi_4+\phi_x) & \sin(\phi_4+\phi_x) \\ \cos(\psi_4+\psi_x)\sin(\theta_4+\theta_x) + \sin(\psi_4+\psi_x)\sin(\phi_4+\phi_x)\cos(\theta_4+\theta_x) & \sin(\psi_4+\psi_x)\sin(\theta_4+\theta_x) - \cos(\psi_4+\psi_x)\sin(\phi_4+\phi_x)\cos(\theta_4+5ta_x) & \cos(\phi_4+\phi_x)\cos(\theta_4+\theta_x) \end{bmatrix} \begin{bmatrix} X_Z \\ Y_Z \\ Z_Z \end{bmatrix} \tag{4}$$

The sunlight azimuth and pitch angles in the guide mirror coordinate system are shown in Figure 5; these are the rotation angles around the $Z$ axis and $X$ axis, respectively. The sunlight is incident from the $-Y$ axis, and the solar vector is the normalized value. Therefore, the conditions for calculating the azimuth and pitch angles are: $-32767 < Y_J < 0$ and $32767 > Z_J > -32767$. The rotation angle of the turntable adopts the left-hand rule. In order to ensure that the solar-tracking simulation experiment on the ground is sufficient to avoid incorrect angle value symbols, the SECCS can rectify the signs of the sun angle according to the injection parameters received in orbit. Thus, the azimuth angle ($\delta$) and pitch angle ($\eta$) are calculated from the solar vector in the guide mirror coordinate system while rectifying the deviation and signs, as shown in Formulas (5) and (6):

$$\delta = \lambda_a * \text{arctg} \frac{X_J}{Y_J} \tag{5}$$

$$\eta = \lambda_{\mathrm{P}} * \mathrm{arcsin} \frac{Z_J}{32767} \tag{6}$$

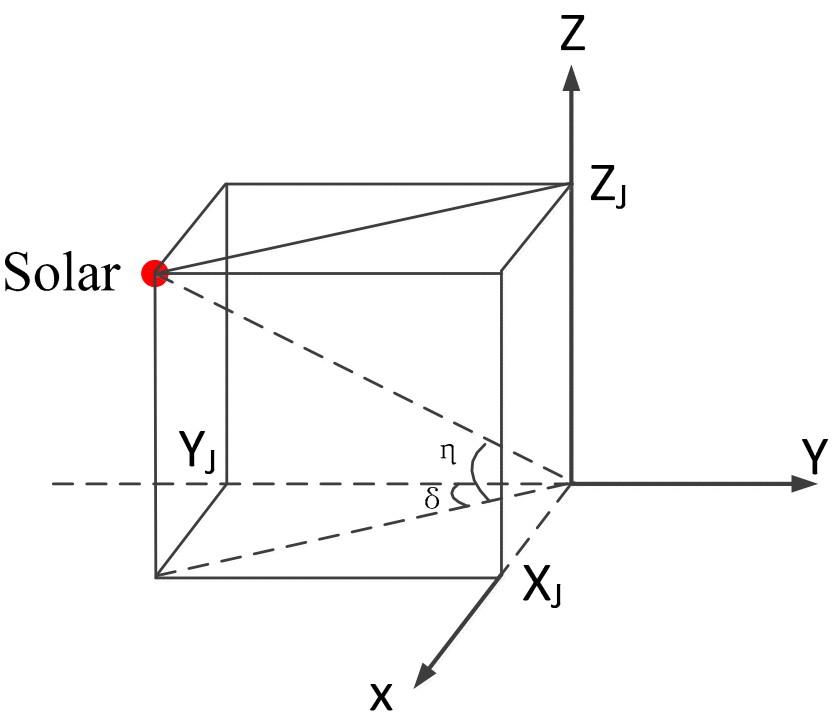

**Figure 5.** Sunlight azimuth and pitch angle in the guide-mirror coordinate system.

The calculation and rectification of sunlight angles are very complex. When the solar-tracking simulation experiment on the ground is not sufficient, resulting in an inaccurate angle calculation, the SECCS dose not rectify the solar vector with Formulas (1)–(4) according to the injection instructions received in orbit, and the angle is directly calculated from the solar vector under the orbital coordinate system. The azimuth and pitch angle calculation processes, with sign rectification only, are shown in Formulas (7) and (8).

$$\delta = \lambda_{\mathrm{a}} * \mathrm{arctg} \frac{X_S}{Y_S} \tag{7}$$

$$\eta = \lambda_{\mathrm{p}} * \mathrm{arcsin} \frac{Z_S}{32767} \tag{8}$$

$\lambda_{\mathrm{a}}$ and $\lambda_{\mathrm{p}}$ are the sign-correction parameters of the solar azimuth and pitch angles during injection, and the value is +1 or −1.

## 3. Results of High-Precision Solar Tracking in Orbit

The spectrometer carried out two experiments using the alternative method for sunlight tracking in orbit. One tracking experiment using the primary method was chosen to analyze the quality of in-orbit sunlight tracking. The sunlight angle, the working-mode switch, the guide mirror's offset, and the guide mirror's offset during the detection process when using the alternative method are shown in Figures 6 and 7. The same parameters for the primary method are shown in Figure 8. The precision of the sun position calculation is validated with the guide mirror's offset during the detection process. In future research, the combination of the two methods can be considered to improve accuracy.

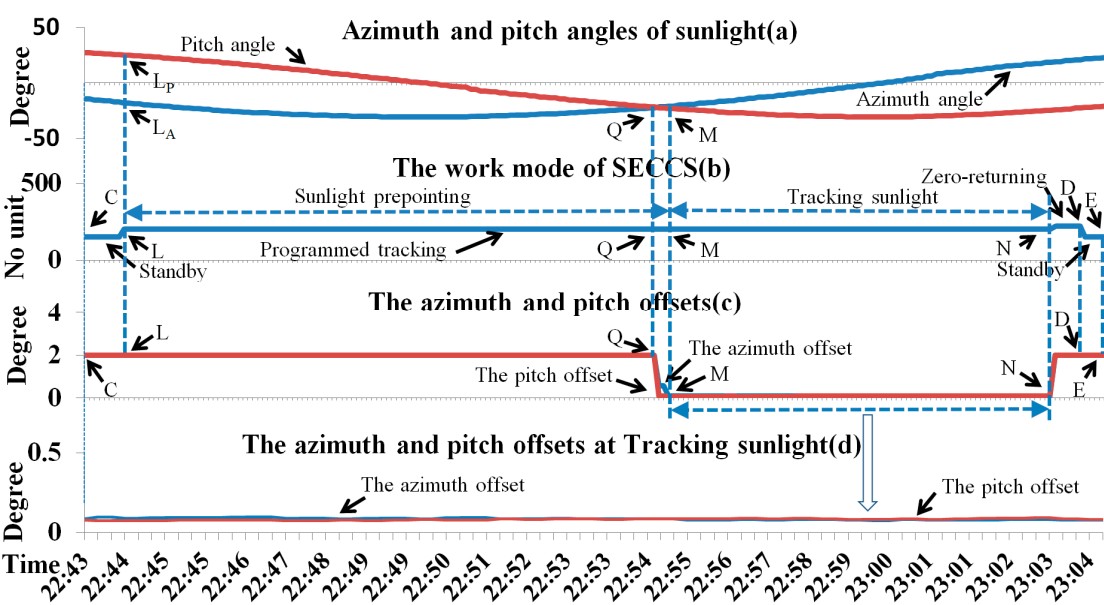

**Figure 6.** The work mode and angle in the first orbit using the alternative method.

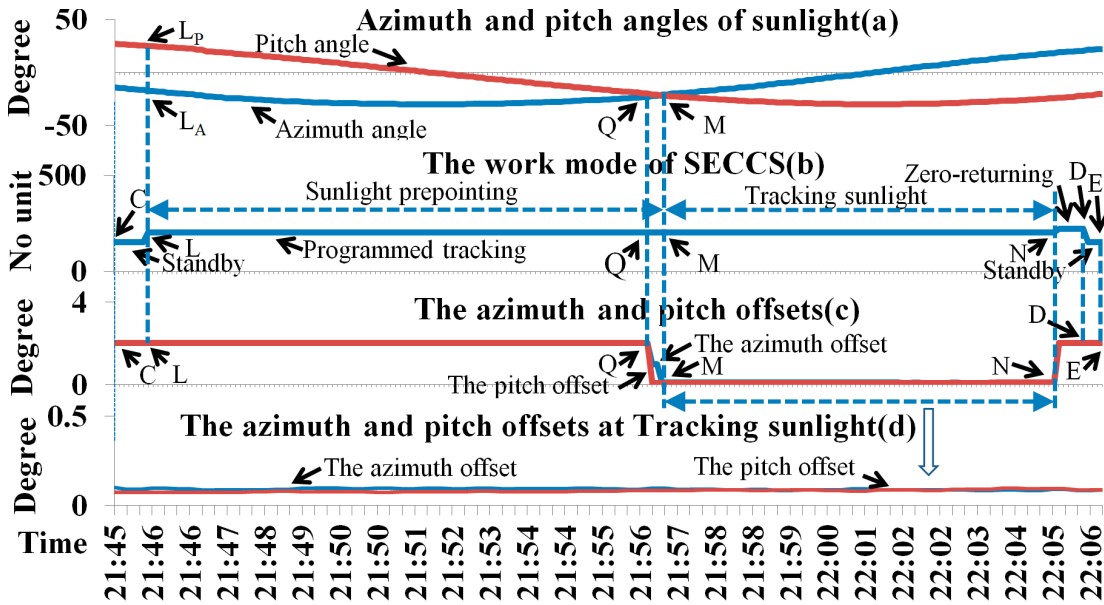

**Figure 7.** The work mode and angle in the second orbit using the alternative method.

The rotation range of the turntable azimuth direction was changed from −30°~24° to −21°~24° according to the injection parameters, and the rotation range of the pitch direction was −34.5°~0°. When the sunlight was outside the field of view of the guide mirror, the offset returned by the mirror was a fixed value of 2°. All data readings transmitted to the ground cycle was 32 s.

We take Figure 6 as an example to analyze the system operation under the conditions of the alternative method.

The spectrometer's electrical cabinet controlled the turntable to be in the standby state at the CL section, as shown in curve b. The sunlight was outside the guide mirror's field of view, and the azimuth and pitch offsets were 2°, as shown in curve c. The working mode was switched to the programmed tracking (solar prepointing).

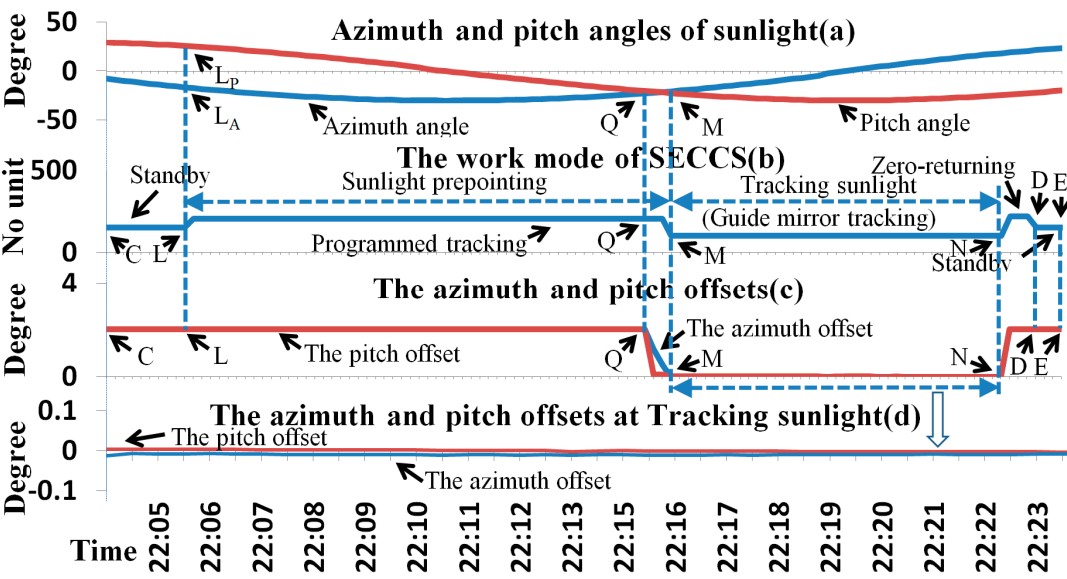

**Figure 8.** The work mode and angle using the primary method.

When the received orbit message count reached a certain value, the working mode was switched to programmed tracking (solar prepointing) at the LQ section, as shown in curve b. The sunlight azimuth and pitch angles were −22.226° and −21.568°, respectively, at point Q of curve a. The sunlight was outside the rotation range of the turntable at the LQ segment. Additionally, it was outside the guide mirror's field of view, and its returned azimuth and pitch offsets were 2°, as seen in the LQ section of curve c. The sunlight was in the field of view of the guide mirror at point Q.

The sunlight azimuth and pitch angles were −20.783° and −22.959°, respectively, at point M of curve a. The sunlight was in the field of view of the guide mirror at the QM segment. The sunlight azimuth and pitch offsets were true values at the QM segment of curve c. Because the change rate of the sunlight azimuth and pitch angles at the PM segment were different, the change in the guide mirror's offset in the corresponding direction was also different. At that time, the sunlight was still outside the field of view of the guide mirror; the working mode was programmed tracking (solar prepointing), as seen in curve b. At point M, the sunlight entered the rotating range of the turntable.

The SIS started to track and detect the sun at point M and finished doing so at point N of curve a, where the azimuth angle and the picking angle were 18.677° and −24.661°, respectively. The sunlight was within the rotation range of the turntable, and the working mode was programmed tracking (sunlight tracking) at the MN section of curve b. The azimuth and pitch offsets of the guide mirror are shown in curve d at the MN section; they were within the range of 0.074~0.098°. The working mode was about to enter the zero-return mode at point N of curve b.

The working mode was the zero-return mode in the ND section of curve b, and the turntable completed the zero-return action at point D. The sunlight gradually moved out of the guide mirror's field of view in the ND segment, and the changes of azimuth and pitch offset of the guide mirror are shown in the ND segment of curve c. The working mode switched to standby at point D.

The working mode was standby in the DE segment of curve b; the sunlight moved out of the guide mirror's field of view, and the azimuth and pitch deviation are shown in curve c. Here, the solar irradiance spectrometer waited for the next sunlight tracking and detection mission.

The mode switch, the change in the guide mirror offset, and the timing of the sunlight detection in Figure 7 are the same as in Figure 6, so we do not repeat them. The deviation of the guide mirror during the sunlight detection process (MN segment) is shown in curve d in Figure 7, and the azimuth and pitch deviation range was 0.071~0.0992°.

The sunlight angle, the working-mode switch, the offset of the guide mirror, and the offset of the guide mirror during in-orbit sunlight detection (MN segment) with the primary method are shown in Figure 8. Compared with Figures 6 and 7, the working mode in Figure 8 was different only in the MN segment. During that period, the working mode was guide-mirror tracking, as shown in curve b, and the range of the azimuth and pitch offset was −0.0121~0.0037°.

Since the data update cycle was 32 s, the solar angle curves were not smooth.

Figure 9 shows the solar spectral curves under the following conditions: curves a and c correspond to the first and second orbit using the alternative method; curve b shows the orbit when using the primary method. These curves measured in the ultraviolet to near-infrared bands were consistent; there was a slight deviation between them, but this did not affect the data inversion.

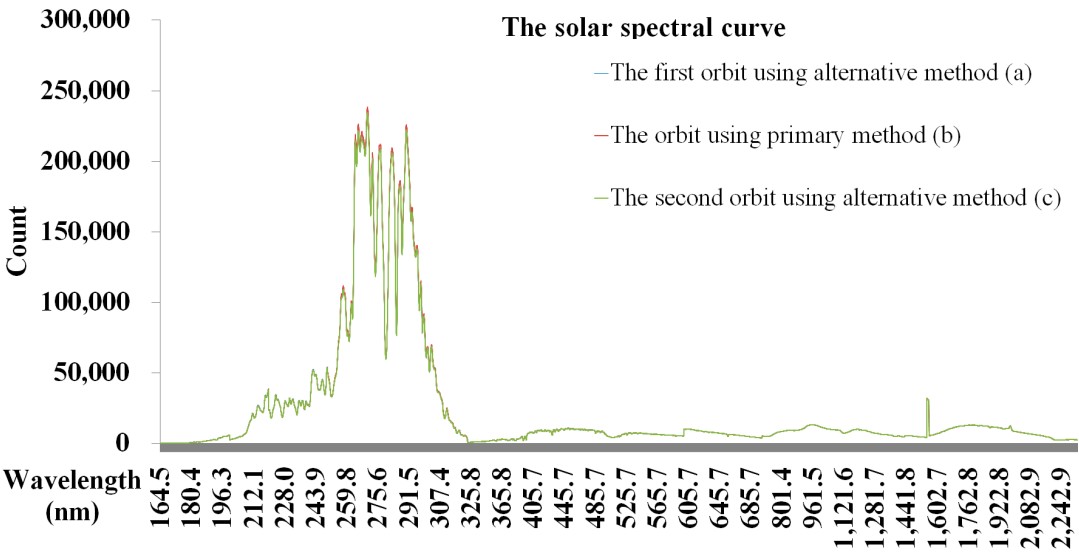

**Figure 9.** The solar spectral curve.

## 4. Discussion

The two methods of solar tracking and the in-orbit detection show the following:

1. The rotation range of the turntable azimuth direction changes in orbit;
2. The SECCS calculates the sunlight angle with Formula (4); $\psi_x$, $\varphi_x$, and $\theta_x$ are 0° in orbit;
3. The mode switches in orbit are the same as the simulation on the ground;
4. The sunlight tracking accuracy meets the requirements of remote sensing instruments, and the solar spectral detection results are stable in orbit.

Contents to be further verified include the following:

1. All of the azimuth and pitch offsets of curve d are positive in Figures 6 and 7 in orbit.
2. The relationship between the guide mirror's offset values and the count of the solar spectral irradiance is uncertain.

## 5. Conclusions

Reliability is the most important factor in space missions, and tracking moving targets is the most important function of remote sensing instruments. Herein, two comprehensive methods for capturing and tracking the solar position were strenuously designed; they can be implemented by software simply and reliably. The results showed that these methods were successful in orbit.

The error and deviation of the solar tracking system were analyzed in detail, which demonstrated the feasibility of the system design. The correctable deviations were identified and a correction method was proposed, and the software tool was optimized to correct the

mechanical installation deviation. The software tool calculated the solar angle value with high precision when a calibration was adopted. It was found that the tracking precision of the alternative method was better than $0.9960°$; that method met the tracking requirements. The primary method's tracking precision was better than $0.0121°$. This enabled remote sensing instruments to detect the solar position in the ultraviolet to near-infrared spectrum in orbit stably and with high precision.

## 6. Patents

We have applied for Chinese patents for our method, under the patent name of "a method of sunlight angle calculation and rectification in orbit". The patent application number is 202211501766.9. It is currently awaiting review by the China National Intellectual Property Administration.

**Author Contributions:** Methodology, Y.S.; Software, Y.S.; Data curation, Y.S., Z.L., X.Y., Y.H., B.L., G.L. and J.L.; Supervision, Y.S. All authors have read and agreed to the published version of the manuscript.

**Funding:** This research was funded by National Natural Science Foundation of China (No. 62205330) and National key research and development program of China (No. 2022YFB3903202).

**Data Availability Statement:** The data is unavailable due to privacy or ethical restrictions.

**Conflicts of Interest:** The authors declare no conflict of interest.

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
