# Peer review of "Methods of Analyzing the Error and Rectifying the Calibration of a Solar Tracking System for High-Precision Solar Tracking in Orbit"

_remotesensing, doi:10.3390/rs15082213_

Round 1
Reviewer 1 Report
one of the method is related to the sun position by calculation , it should be give the detailed calculation equation for the sun position. it can be validated with a more accurate method. please refer to,
Applied Sciences | Free Full-Text | Evaluation and Characterization of the Influence of Solar Position Algorithm on the Performance of Parabolic Trough Solar System https://www.mdpi.com/2076-3417/13/3/1821
Author Response
Dear specialist,
Thanks very much for your appreciated advice.The manuscript (remotesensing-2312121) was revised based on your advice.The modification should be achieved in PDF file.

Reviewer 2 Report
The authors proposed a special strategy for determining injection parameters aimed at successfully capturing and tracking sunlight even with vibration and mechanical displacement of structures as a result of launch. In addition, the described methods can be applied to orbits with any inclination angles. However, before publication, the authors should carefully correct some inaccuracies in the text that make it difficult to understand their methodology.
However, before publication, the authors should carefully correct some inaccuracies in the text that make it difficult to understand their methodology. For example, in numbered formulas (1 -3) and in the text of the article, the angle (rolling angle) is denoted differently. Understanding some diagrams is difficult because they contain small, poorly readable text that does not fit into algorithmic blocks. Not all designations are introduced immediately before or after their first use, the reader has to search for them throughout the article. The formulas in the text of the article apparently contain errors. For example, in line 243, the formula probably means the sum of the squares of expressions in parentheses, and not the terms multiplied by two. There are quite a lot of such errors.
Author Response

(The authors gave the same response as above.)

Reviewer 3 Report
Remote sensing 27-3-2023
This study presented an interesting topic. However, I have some remarks on this work:
1. The abstract is preferred to include the most significant quantitative results of this work.
2. The literature review needs to be enriched by adding more related sources.
3. Is Figure 2-b a flowchart? If so, the contents of the blocks need to modify especially the decision blocks. Where are the Start, input, output, and End blocks?
4. The reference to Figure 1-a should be mentioned in the figure’s caption unless it's original.
5. A drawing to demonstrate the location of the angles is required for the transformation from the orbital coordinate system to the satellite coordinate system that is associated with equation (1).
6. It is difficult to check the results of this manuscript as the figures from 6 to 9 are unclear. Please enlarge the text fonts for all the result figures. Figure labels should be crisp and easy to read.
Author Response

(The authors gave the same response as above.)

Round 2
Reviewer 1 Report
no--
Reviewer 3 Report
Thanks to the authors to consider all the reviewer comments